# The Convergence of a Cooperation Markov Decision Process System

**DOI:** 10.3390/e22090955

**Published:** 2020-08-30

**Authors:** Xiaoling Mo, Daoyun Xu, Zufeng Fu

**Affiliations:** 1College of Computer Science and Technology, Guizhou University, Guiyang 550025, China; gs.xlmo19@gzu.edu.cn (X.M.); fuzufeng@outlook.com (Z.F.); 2Department of Electronics and Information Engineering, Anshun University, Anshun 561000, China

**Keywords:** reinforcement learning, cooperation markov decision process, multi-agent, optimal pair of strategies

## Abstract

In a general Markov decision progress system, only one agent’s learning evolution is considered. However, considering the learning evolution of a single agent in many problems has some limitations, more and more applications involve multi-agent. There are two types of cooperation, game environment among multi-agent. Therefore, this paper introduces a Cooperation Markov Decision Process (CMDP) system with two agents, which is suitable for the learning evolution of cooperative decision between two agents. It is further found that the value function in the CMDP system also converges in the end, and the convergence value is independent of the choice of the value of the initial value function. This paper presents an algorithm for finding the optimal strategy pair $\left(\pi_{k}^{0},\pi_{k}^{1}\right)$ in the CMDP system, whose fundamental task is to find an optimal strategy pair and form an evolutionary system $CMDP^{\;(\pi_{k}^{0},\pi_{k}^{1})}$. Finally, an example is given to support the theoretical results.

## 1. Introduction

Artificial intelligence technology has become one of the most important technologies, nowadays. AlphaGo, unmanned driving, voice recognition, face recognition and other well-known technologies involve artificial intelligence. Artificial intelligence technology has been widely used in economic construction, education, medical treatment, social life and other fields. Many industries have also begin to evolve to artificial intelligence, intelligence has become an important direction of industrial development. Machine learning generally includes supervised learning, unsupervised learning and reinforcement learning [1]. Among them, reinforcement learning is an important research field. Reinforcement learning is learned by interacting with a dynamic environment through trial and error, is an important branch in the field of machine learning and artificial intelligence. It is leap to control the behavior of agents through rewards and punishments. Reinforcement learning is widely used in many fields, such as: electronic game [2], board games [3], maze game, recommendation system [4], and so forth.

The important basis of reinforcement learning [5] is Markov Decision Process (MDP) system [6]. From the evolution process of the computational model, Markov Process (MP) system is the basic model, which introduces reward function and execution behavior, and introduces a mapping from state set to behavior set as a strategy to solve Markov decision problem. Under the given policy π, the Markov decision process system degenerates into a Markov process system with a reward function $MDP{\;}^{\pi }$. Once the policy is given, the main task of the Markov process system with the reward function is to evolve and generate the expected reward values collected in each state. The main objective of the Markov decision system is to find an optimal strategy [7,8] $\pi^{*}$, so that the expected reward value collected on the Markov chain can be maximized under this strategy.

In the usual Markov decision process system, only an agent is learning and evolving. However, in real problems, the decision-making ability of an agent is far from enough, and many problems can be solved by using a multi-agent [9,10,11]. Therefore, we encounter two kinds of problems: one is that multi-agent restrict each other to accomplish their target tasks in the same environment. Such systems are called antagonistic systems [12] or game systems [13]. In this kind of system, all the intelligent agent executes the strategy behavior alone, according to mutually restricts or plays games with each other intelligent agent to finishes its own target task. The optimal strategy is represented as agent strategy vector group and the reward value is reflected as the reward value of its own execution. Littman [12] proposed minimax-q algorithm for multi-agent competitive reinforcement learning based on Markov decision framework based on zero-sum game in 1994. The other is that multi-agent work together to accomplish a target task in the same environment. Such systems are called collaborative systems [14] or cooperative [15]. In this kind of system, all the intelligence agent executes the strategy behavior alone, according to cooperates with each other to completes one target task jointly. The optimal strategy is the vector set of agent strategy, and the reward value is the social comprehensive value of combinatorial execution. Littman [16] also proposed a collaborative team q-learning algorithm in 2001. The multi-agent cooperative algorithm is used to solve problem about traffic signal signal coordination control [17], multi-agent path planning [18], multi-satellite collaborative task planning [19], and so forth. In the case of multi-agent, the state shift is the result of all agents acting together, so the rewards for agents also depend on the joint strategy. Iwata K et al. [6] analyzed the statistical properties of multi-agent learning based on Markov decision process. However, the multiple agents in this paper are independent, and the joint action will make the state transfer and reward value of the whole system, but only migrates the Markov decision process of a single agent to multi-agent. Zhang W et al. [20] has began to introduce the communication mechanism into the multi-agent system. Although agents begin to communicate with each other, the consumption of memory was enlarged, and agents could not predict the impact of their actions on the action execution of other agents.

This paper only considers the Cooperation Markov Decision Process (CMDP) system of two agents, which is suitable for the evolutionary learning system of cooperative decision between two agents. Two-agent games for multi-agent reinforcement learning are similar to perceptrons for neural networks.In this kind of learning model, agents alternately execute behaviors, seek optimal criteria based on social value, seek optimal strategies $\left(\pi_{k}^{0},\pi_{k}^{1}\right)$, and jointly complete the target task. This paper introduces a cooperation Markov decision process system in the form of definition, two trade agent (Alice and Bob) on the basis of its strategy to perform an action. that is, after Bob observes that Alice performs an action, Bob is deciding which action to perform, and further Bob’s execution of the action will also affect the execution of Alice’s next action. The Markov system learns evolutionally through a given strategy pair and looking for a pair of optimal strategy $\left(\pi_{k}^{0},\pi_{k}^{1}\right)$, form an evolution system $CMDP^{\left(\pi_{k}^{0},\pi_{k}^{1}\right)}$, and gives the algorithm [21,22] to search for the optimal strategy pair. In this paper, the convergence property of the value function of the MDP system with the participation of a single agent is given, the convergence phenomenon of the value function in the cooperation Markov decision process system proposed in this paper is further explored, and the correctness of the property is proved from both the experimental and theoretical perspectives.

## 2. Markov Reward Process System

A finite Markov process system is constituted by two tuples <S,P>, $S=\{s_{1},\cdots \cdots ,s_{n}\}$ is a set of finite states (containing *n* states), define a probability distribution *P* over *S*, $P(s,s^{\prime })$ represents the probability of transition from state *s* to state $s^{\prime }$, what’s more, $\forall s\in S:\Sigma_{s^{\prime }\in S}P\left(s,s^{\prime }\right)=1$. Markov Reward Process is a kind of Markov chain with value, which consisted of four tuples <S,P,R,γ>. Introduced a reward function, $R:\;\;\;S\times S\to R^{+}$ ($R^{+}$ means non-negative real number set). So $R(s,s^{\prime })$ is the reward value obtained by transferring from state *s* to state $s^{\prime }$; 0<γ<1 is the attenuation factor, also known as the discount factor. Recursively defines an expected reward value (value function) collected by Markov process system:(1)$$V\left(s\right)=\Sigma_{s^{\prime }\in S}P\left(s,s^{\prime }\right)\left[R\left(s,s^{\prime }\right)+\gamma \cdot V\left(s^{\prime }\right)\right]\;.$$

Equation (Equation 1) recorded the reward value obtained on the Markov chain $X_{0},X_{1},X_{2},\cdots ,X_{t},X_{t+1},\cdots$ with the state *s* as the initial state. The term $R\left(s,s^{\prime }\right)+\gamma \cdot V\left(s^{\prime }\right)\;$ indicated that: starting from the state *s*, the reward value obtained by one-step transition to $s^{\prime }$ and plus the discount of the expected reward value collected from the state $s^{\prime }$. Equivalently, writen (1) as the following iterative formula:(2)$$V_{k+1}\vec{\left(s\right)}=diag\left(PR^{T}\right)+\gamma \cdot PV_{k}\vec{\left(s\right)}.$$

Among them, $R^{T}$ represents the transpose of the matrix *R* and diag(A) represents the column vector formed by the diagonal elements of the matrix *A*.

Now we make an experiment, experiment 1: take probability matrix *P*, reward function *R*, and discount factor γ as follows. Observe the convergence of the value function *V* through an example.
$$P=\left(\begin{array}{ccccc} 1 & 0 & 0 & 0 & 0\\ 1/3 & 0 & 1/3 & 0 & 1/3\\ 0 & 0 & 0 & 1/3 & 2/3\\ 1/2 & 0 & 0 & 1/2 & 0\\ 0 & 1/4 & 0 & 3/4 & 0 \end{array}\right)R=\left(\begin{array}{ccccc} 2 & 0 & 0 & 0 & 0\\ 2 & 0 & 2 & 0 & 3\\ 0 & 0 & 0 & 1 & 2\\ 1 & 0 & 0 & 2 & 0\\ 0 & 1 & 0 & 2 & 0 \end{array}\right)\gamma =0.85$$

Initially, initial vector is set as $V_{0}={\left[0,0,0,0,0\right]}^{T}$, the convergence of the value function is shown in the following Figure 1:

According to Figure 1, we can observed that starting from the initial state, the value function (*V*-function) will eventually converge as the number of iterations steps. We further speculate that the final convergence value has nothing to do with the initial value setting.

**Theorem** **1.**
*Under any initial state $V_{0}\vec{\left(s\right)}$ , randomly walk on the Markov reward process system, the value function $V_{k}\vec{\left(s\right)}$ finally obtain converges.*


**Proof.** From the iterative Formula (2), it is easy to obtain that,
(3)$$\begin{array}{c} V_{k}\vec{\left(s\right)}=diag\left(PR^{T}\right)+\gamma \cdot PV_{k-1}\vec{\left(s\right)};\;V_{k-1}\vec{\left(s\right)}=diag\left(PR^{T}\right)+\gamma \cdot PV_{k-2}\vec{\left(s\right)};\\ V_{2}\vec{\left(s\right)}=diag\left(PR^{T}\right)+\gamma \cdot PV_{1}\vec{\left(s\right)};\;V_{1}\vec{\left(s\right)}=diag\left(PR^{T}\right)+\gamma \cdot PV_{0}\vec{\left(s\right)}; \end{array}$$
simplify Equation (Equation 3) to calculate the difference in value function between (k+1)-step and *k*-step,
(4)$$\begin{array}{c} V_{k+1}\vec{\left(s\right)}-V_{k}\vec{\left(s\right)}=\gamma \cdot P\left(V_{k}\vec{\left(s\right)}-V_{k-1}\vec{\left(s\right)}\right)=\gamma^{2}\cdot P^{2}\left(V_{k-1}\vec{\left(s\right)}-V_{k-2}\vec{\left(s\right)}\right)\\ \;\;\;\;\;=\gamma^{3}\cdot P^{3}\left(V_{k-2}\vec{\left(s\right)}-V_{k-3}\vec{\left(s\right)}\right)=\gamma^{k}\cdot P^{k}\left(V_{1}\vec{\left(s\right)}-V_{0}\vec{\left(s\right)}\right)\\ \;\;\;\;\;=\gamma^{k}\cdot P^{k}\cdot diag\left(PR^{T}\right)+\gamma^{k}\cdot P^{k}\cdot \left(\gamma \cdot P-E\right)V_{0}\vec{\left(s\right)}. \end{array}$$Let *P* is the probability transfer matrix, *R* is the reward value matrix, the discount factor satisfy 0<γ<1. We have
(5)$$\begin{array}{c} {\left[\begin{array}{ccc} R_{\min}^{1} & \cdots & R_{\min}^{n} \end{array}\right]}^{T}-\left(V_{\max}-\gamma \cdot V_{\min}\right){\left[\begin{array}{ccc} 1 & \cdots & 1 \end{array}\right]}^{T}\leq diag\left(PR^{T}\right)-\left(E-\gamma \cdot P\right)V_{0}\vec{\left(s\right)}\\ \leq {\left[\begin{array}{cccc} R_{\max}^{1} & R_{\max}^{2} & \cdots \; & R_{\max}^{n} \end{array}\right]}^{T}-\left(V_{\min}-\gamma \cdot V_{\max}\right){\left[\begin{array}{cccc} 1 & 1 & \cdots & 1 \end{array}\right]}^{T}, \end{array}$$
where $R_{\min}^{i}$ and $R_{\max}^{i}$ record the minimum and maximum value of the reward value in the *i*-th row of the reward matrix *R*, respectively. $V_{0}\vec{\left(s\right)}$ represent the value of the initial moment: $V_{0}\vec{\left(s\right)}={\left[\begin{array}{cccc} V_{01} & V_{02} & \cdots & V_{0n} \end{array}\right]}^{T}$, the minimum and maximum value of *V* are $V_{\min}=\min\left\{V_{0}\vec{\left(s\right)}\right\}$ and $V_{\max}=\max\left\{V_{0}\vec{\left(s\right)}\right\}$, respectively. Reduce the left side of the inequality (2.5) and enlarge the right side. Then we conclude,
$$\begin{array}{c} \gamma^{k}\cdot \left(R_{\min}^{*}-V_{\max}+\gamma \cdot V_{\min}\right){\left[\begin{array}{cccc} 1 & 1 & \cdots & 1 \end{array}\right]}^{T}\leq V_{k+1}\vec{\left(s\right)}-V_{k}\vec{\left(s\right)}\\ V_{k+1}\vec{\left(s\right)}-V_{k}\vec{\left(s\right)}\leq \gamma^{k}\cdot \left(R_{\max}^{*}-V_{\min}+\gamma \cdot V_{\max}\right){\left[\begin{array}{cccc} 1 & 1 & \cdots & 1 \end{array}\right]}^{T}, \end{array}$$
where $R_{\min}^{*}$ and $R_{\max}^{*}$ record the minimum and maximum value in the reward matrix *R*, respectively.Finally, because of the discount factor 0<γ<1, with the constant increase of *k*, the value of $\gamma^{k}$ approaches zero infinitely, so the influence of $R_{\min}^{*}$, $R_{\max}^{*}$, $V_{\max}$ and $V_{\min}$ are much smaller than $\gamma^{k}$ under general constants number, therefore $0\leq V_{k+1}\vec{\left(s\right)}-V_{k}\vec{\left(s\right)}\leq 0$. Obviously, when the value of *k* is large, $V_{k+1}\vec{\left(s\right)}=V_{k}\vec{\left(s\right)}$, that is to say the value of the final value function converges. □

**Theorem** **2.**
*The choice of initial value $V_{0}\vec{\left(s\right)}$ does not affect the convergence value of the final value function $V_{k}\vec{\left(s\right)}$.*


**Proof.** Simplifying Equation (Equation 3):
(6)$$\begin{array}{c} V_{k}\vec{\left(s\right)}=diag\left(PR^{T}\right)+\gamma \cdot PV_{k-1}\vec{\left(s\right)}=diag\left(PR^{T}\right)+\gamma \cdot P\cdot \left(diag\left(PR^{T}\right)+\gamma \cdot PV_{k-2}\vec{\left(s\right)}\right)\\ \;\;\;=diag\left(PR^{T}\right)+\gamma \cdot P\cdot \left(diag\left(PR^{T}\right)+\gamma \cdot P\cdot \left(diag\left(PR^{T}\right)+\gamma \cdot PV_{k-3}\vec{\left(s\right)}\right)\right)\\ \;\;\;=\left(E+\gamma \cdot P+{\left(\gamma \cdot P\right)}^{2}+{\left(\gamma \cdot P\right)}^{3}+...+{\left(\gamma \cdot P\right)}^{k-1}\right)\cdot diag\left(PR^{T}\right)+{\left(\gamma \cdot P\right)}^{k}\cdot V_{0}\vec{\left(s\right)}. \end{array}$$Because *P* is a probability transition matrix, then $P^{k}$ is a probability transition matrix also. Then we can get,
(7)$$\left[\begin{array}{c} R_{\min}^{*}\\ R_{\min}^{*}\\ \vdots \\ R_{\min}^{*} \end{array}\right]\leq \left[\begin{array}{c} R_{\min}^{1}\\ R_{\min}^{2}\\ \vdots \\ R_{\min}^{n} \end{array}\right]\leq p^{i}\cdot diag\left(PR^{T}\right)\leq \left[\begin{array}{c} R_{\max}^{1}\\ R_{\max}^{2}\\ \vdots \\ R_{\max}^{n} \end{array}\right]\leq \left[\begin{array}{c} R_{\max}^{*}\\ R_{\max}^{*}\\ \vdots \\ R_{\max}^{*} \end{array}\right],i\in \left\{1,2,\cdots ,k\right\},$$
(8)$${\left[\begin{array}{cccc} V_{\min} & V_{\min} & \cdots \; & V_{\min} \end{array}\right]}^{T}\leq P^{k}\cdot V_{0}\vec{\left(s\right)}\leq {\left[\begin{array}{cccc} V_{\max} & V_{\max} & \cdots & V_{\max} \end{array}\right]}^{T},$$
applying (7), (8) and simplifying gives
(9)$$V_{k}\vec{\left(s\right)}\geq (1+\gamma +\gamma^{2}+\cdots +\gamma^{k-1})\cdot R_{\min}^{*}+\gamma^{k}\cdot V_{\min},$$
(10)$$V_{k}\vec{\left(s\right)}\leq (1+\gamma +\gamma^{2}+\cdots +\gamma^{k-1})\cdot R_{\max}^{*}+\gamma^{k}\cdot V_{\max},$$
expand the inequality (9),
(11)$$\begin{array}{c} (1+\gamma +\gamma^{2}+\cdots +\gamma^{k-1})\cdot R_{\min}^{*}+\gamma^{k}\cdot V_{\min}=\left(\frac{1}{\gamma^{k}}+\frac{1}{\gamma^{k-1}}+\cdots +\frac{1}{\gamma }\right)\cdot \gamma^{k}\cdot R_{\min}^{*}+\gamma^{k}\cdot V_{\min}\\ =\gamma^{k}\left[\left(\frac{1}{\gamma^{k}}+\frac{1}{\gamma^{k-1}}+\cdots +\frac{1}{\gamma }\right)\cdot R_{\min}^{*}+V_{\min}\right]\approx \gamma^{k}\left[\left(\frac{1}{\gamma^{k}}+\frac{1}{\gamma^{k-1}}+\cdots +\frac{1}{\gamma }\right)\cdot R_{\min}^{*}\right]; \end{array}$$
expand the inequality (10),
(12)$$\begin{array}{c} (1+\gamma +\gamma^{2}+\cdots +\gamma^{k-1})\cdot R_{\max}^{*}+\gamma^{k}\cdot V_{\max}=\left(\frac{1}{\gamma^{k}}+\frac{1}{\gamma^{k-1}}+\cdots +\frac{1}{\gamma }\right)\cdot \gamma^{k}\cdot R_{\max}^{*}+\gamma^{k}\cdot V_{\max}\\ =\gamma^{k}\left[\left(\frac{1}{\gamma^{k}}+\frac{1}{\gamma^{k-1}}+\cdots +\frac{1}{\gamma }\right)\cdot R_{\max}^{*}+V_{\max}\right]\approx \gamma^{k}\left[\left(\frac{1}{\gamma^{k}}+\frac{1}{\gamma^{k-1}}+\cdots +\frac{1}{\gamma }\right)\cdot R_{\max}^{*}\right]. \end{array}$$It is found that, because of the discount factor 0<γ<1, the value of $\frac{1}{\gamma^{k}}$ continuously approaches infinity as the value of *k* increases, so $\frac{1}{\gamma^{k}}+\frac{1}{\gamma^{k-1}}+\cdots +\frac{1}{\gamma }\gg 1$ and monotonically increase on the Formulas (11) and (12). Therefore, the effects of $V_{\min}$ and $V_{\max}$ in the initial value function $V_{0}\vec{\left(s\right)}$ can be ignore for the final value function $V_{k}\vec{\left(s\right)}$. so the choice of the initial $V_{0}\vec{\left(s\right)}$ does not affect the convergence value of the final value function $V_{k}\vec{\left(s\right)}$ . It further proves that our conjecture is correct. □

## 3. Markov Decision Process System

Markov decision process system introduces the behavior in Markov reward process system and modifies the reward function to the behavior execution step. MDP system is a random process on state, reward, action sequences, and consists of five-tuples <S,A,P,R,γ>. $A=\{a_{1},...,a_{m}\}$ is a set of finite action, $P(s,a,s^{\prime })$ represents the probability of transition from state *s* to state $s^{\prime }$ by performing action *a*. $P_{}:S\times A\times S\to \left[0,1\right]$, $P\left(s,a,s^{\prime }\right)=P\left(s_{t+1}=s^{\prime }|s_{t}=s^{\prime },a_{t}=a\right)$, where for each pair of "state-behavior" (s,a), $\Sigma_{s^{\prime }\in S}P\left(s,a,s^{\prime }\right)=1$. The reward function *R* define with, $R_{}:S\times A\times S\to R$, where $R\left(s,a,s^{\prime }\right)=E\left(R_{t+1}|s_{t}=s,a_{t}=a\right)$. Similarly, γ is the attention factor also called discount factor, satisfy 0<γ<1. π is a strategy of MDP system, π:S→A. Such as, π(s)=a represents the execution of action *a* in state *s*. So, let $V^{\pi }\left(s\right)$ is the value of *s* under a given policy π, we can include
(13)$$V^{\pi }\left(s\right)=\Sigma_{s^{\prime }\in S}^{}P\left(s,\pi \left(s\right),s^{\prime }\right)\left[R\left(s,\pi \left(s\right),s^{\prime }\right)+\gamma \cdot V^{\pi }\left(s^{\prime }\right)\right]\;\;\;.$$

Equation (Equation 13) is called the Bellman recursive equation. In the function $V^{\pi }$, if see behavior as a variable, we can induce a state-action value function, that is, a *Q*-function, Q:S×A→R
(14)$$Q^{\pi }\left(s,a\right)=\Sigma_{s^{\prime }\in S}^{}P\left(s,a,s^{\prime }\right)\left[R\left(s,a,s^{\prime }\right)+\gamma \cdot V^{\pi }\left(s^{\prime }\right)\right]\;.$$

$Q^{\pi }\left(s,a\right)\;$ is understood as follows: starting from state *s*, after executing action *a* and following executing strategy π, the expected reward value is collected. The goal for any given Markov decision process is to find an optimal strategy, that is, to obtain the most reward strategy.

Policy function π:S→A, determines the *V*-function and Q-function that indicate the expected collection of reward values, and $Q^{\pi }$ can be determined by $V^{\pi }$. Therefore, our goal is to find a strategy π that maximizes the function $V^{\pi }$. A *V*-function is optional means if for any policy π, for any state s∈S, satisfies $V^{*}\left(s\right)\geq V^{\pi }\left(s\right)$. A policy is optimal indicates if for any policy π, for any state s∈S, satisfies $V^{\pi *}\left(s\right)\geq V^{\pi }\left(s\right)$. So, if $V^{\pi *}$ function is an optional *V*-function $V^{*}$, there is $V^{*}\left(s\right)=V^{\pi *}\left(s\right)$, for any state s∈S.

A natural problem is the existence of optimal strategy $\pi^{*}$. Second, if it exists, how should we to calculate $\pi^{*}$ ?

According to Theorem 1 in Section 2, the value function of the Markov reward process system will converge, and the value function of the Markov decision process system formed by introducing behaviors on Markov reward process system must also converge. Theoretical results [23,24] show that: an optimal strategy for a finite Markov decision process system <S,A,P,R,γ> exists, and the optimal *V*-function $V^{*}$ and the optimal strategy $\pi^{*}$ have the following relationship:(i)If the optimal function $V^{*}$ has been obtained, the optimal strategy $\pi^{*}$ can be obtained:
(15)$$\pi^{*}\left(s\right)=\arg\max_{a}\Sigma_{s^{\prime }\in S}^{}P\left(s,a,s^{\prime }\right)\left[R\left(s,a,s^{\prime }\right)+\gamma \cdot V^{*}\left(s^{\prime }\right)\right]\;\;\;;$$(ii)If the optimal policy $\pi^{*}$ has been obtained, the optimal function $V^{*}$ can be obtained:
(16)$$V^{*}\left(s\right)=V^{\pi *}\left(s\right)=\Sigma_{s^{\prime }\in S}^{}P\left(s,\pi^{*}\left(s\right),s^{\prime }\right)\left[R\left(s,\pi^{*}\left(s\right),s^{\prime }\right)+\gamma \cdot V^{\pi *}\left(s^{\prime }\right)\right]\;\;\;\;.$$

Equations (15) and (16) give a cross-iterative solution process, and finally the optimal strategy $\pi^{*}$ and the optimal function $V^{*}$ are obtained.

The execution process of the MDP system is shown in the following Figure 2:

## 4. Cooperation Markov Decision Process System

This section introduces a joint formal system that two agents run in the same Markov decision process system, called the cooperation Markov decision process system.

Suppose $A={\left(a_{i,j}\right)}_{n\times n},\;\;\;\;B={\left(b_{i,j}\right)}_{m\times m}\;\;$ are two matrices, the tensor product of matrix *A* and *B* is defined as: $A⊗B=(a_{i,j}B)\;$. If $P_{1},\;\;\;P_{2}$ are two probability matrices but not necessarily the same number of rows, so $P_{1}⊗P_{2}$ is a probability matrix.

Assume $S^{1}=\left\{s_{1}^{1},\cdots ,s_{n}^{1}\right\}$, $S^{2}\;=\left\{s_{1}^{2},\cdots ,s_{m}^{2}\right\}$ are two state sets, $\left(S^{1},P_{1}\right),\;\;\;\left(S^{2},P_{2}\right)$ are two MDP systems, state set $S=S^{1}\times S^{2}$, *S*’s state transition matrix is $P=P_{1}⊗P_{2}$, then MDP system (S,P) is the product system of MDP system $\left(S^{1},P_{1}\right)$ and MDP system $\left(S^{2},P_{2}\right)$.

### 4.1. Cooperation Markov Decision Process System with Two
Agents

We now consider that, in the same system, there are two agents *A* and *B*, each agent performing their own action and completing the target task in a cooperative manner. For the “cooperative” type, the two agents cooperate to accomplish the same goal. Formally, such a system is defined as the following quintuple:$$<S,\;A,\;\left\{Agent_{0},Agent_{1}\right\},\;\left\{P_{0},\;\;P_{1}\right\},\;\left\{R_{0},\;\;R_{1}\right\}>$$

Among them, *S* is a finite state set, here represents the state pair of two agents; *A* is an action set, here represents the action pair of two agents; $P_{i}$ is the probability transfer function of $Agent_{i}$, i=(0,1), define as $P_{i}:S\times S\times A\times S\times A\to \left[0,1\right]$. For any $(s_{1},s_{2},a)\in S\times S\times A$, there hold that $\Sigma_{(s^{\prime },a^{\prime })\in S\times A}P_{i}\left(s_{1},s_{2},a,s^{\prime },a^{\prime }\right)=1$. In other words, for any fixed $\left(s_{1},s_{2},a\right)$, the $P_{i}\left(s_{1},s_{2},a,\;\;•\;,\;\;\;\;•\;\;\right)$ specifies a probability distribution for S×A. Intuitively, $P_{0}\left(s_{1},s_{2},a,s^{\prime },a^{\prime }\right)$ express that when $Agent_{0},Agent_{1}$ in state pair $\left(s_{1},s_{2}\right)$, when $Agent_{0}$ after executes action *a* transitions to $s^{\prime }$ state, and the probability that collaborator $Agent_{1}$ responds to execution of action $a^{\prime }$. $R_{i}$ is the reward function of $Agent_{i}$, i=(0,1), defined as: $R_{i}:S\times S\times A\times S\times A\to R$ (real number set). Intuitively, $R_{0}\left(s_{1},s_{2},a,s^{\prime },a^{\prime }\right)$ express that when $Agent_{0}$, $Agent_{1}$ in state pair $\left(s_{1},s_{2}\right)$, $Agent_{0}$ after executes action *a* transitions to $s^{\prime }$ state, and the reward value that collaborator $Agent_{1}$ responds to execution of action $a^{\prime }$.

Obviously, two agents $Agent_{0}$, $Agent_{1}$ correspond to two *V*-functions: $V_{i}:\;\;\;S\times S\to R\;\;\;\;\left(i=0,1\right)$, in the joint form, we use the social value to describe the joint *V*-function V:S×S→R. Formally defined as:(17)$$V\left(s,t\right)=\alpha V_{0}\left(s,t\right)+\left(1-\alpha \right)V_{1}\left(s,t\right),\;\left(0<\alpha <1\right)\;.$$

Among them, parameter α is called balance parameter.

Policy function defined as: $\pi_{i}:\;\;\;S\times S\to A\;\;\;\;\;\left(i=0,1\right)$, $\pi_{i}\left(s,s^{\prime }\right)=a$ understand as: when $Agent_{i}$ is in *s* state and observe that when $Agent_{1-i}$ is in $s^{\prime }$ state, $Agent_{i}$ executes action *a*. For a given strategy pair $\left(\pi_{0},\pi_{1}\right)$, the evolutionary cooperation Markov decision process system $CMDP^{\left(\pi_{0},\pi_{1}\right)}$ obtains a stable *V*-function whose iterative equation is:(18)$$\begin{array}{c} V_{k+1}^{0}\left(s,t\right)=\Sigma_{s^{\prime }\in S}^{}\Sigma_{a\in A}^{}P_{0}\left(s,t,\;\;\;\pi_{0}\left(s,t\right),s^{\prime },\;\;\;\;a\right)\left[R_{0}\left(s,t,\;\;\;\pi_{0}\left(s,t\right),s^{\prime },\;a\right)+\gamma \cdot V_{}^{k}\left(s^{\prime },t\right)\right]\;\;\;\\ V_{k+1}^{1}\left(s,t\right)=\Sigma_{t^{\prime }\in S}^{}\Sigma_{a\in A}^{}P_{1}\left(s,t,\;\;\;\;\pi_{1}\left(s,t\right),\;\;\;\;t^{\prime },\;a\right)\;\;\;\;\left[R_{1}\left(s,t,\;\;\;\pi_{1}\left(s,t\right),t^{\prime },\;\;a\right)+\gamma \cdot V_{}^{k}\left(s,t^{\prime }\right)\right]\;\;\\ V_{k+1}^{}\left(s,t\right)=\alpha V_{k+1}^{0}\left(s,t\right)+\left(1-\alpha \right)V_{k+1}^{1}\left(s,t\right). \end{array}$$

### 4.2. Convergence of the Social Value Function of CMDP System

Since the value function will eventually converge during Markov’s decision-making process, we guess whether the value function also has the same properties in CMDP system. We first observe by an experiment:

Suppose a CMDP system $<S,\;A,\;\left\{Agent_{0},Agent_{1}\right\},\;\left\{P_{0},\;\;P_{1}\right\},\;\left\{R_{0},\;\;R_{1}\right\}>$, where the discount factor γ=0.95, error coefficient setting ε=0.0001, a state set
$$S=\{(s_{0},s_{0}),\left(s_{0},s_{1}\right),\left(s_{0},s_{2}\right),(s_{1},s_{0}),\left(s_{1},s_{1}\right),\left(s_{1},s_{2}\right),(s_{2},s_{0}),\left(s_{2},s_{1}\right),\left(s_{2},s_{2}\right)\},$$
an action set $A=\{(a_{0},a_{0}),(a_{0},a_{1}),(a_{1},a_{0}),(a_{1},a_{1})\}$, the probability transition function *P* and the reward function *R* are as follows:$$P_{1}\left(S,a_{0},S\right)=P_{0}\left(S,a_{0},S\right):\;\;\;\;\;\;\;\;\;\;\;\begin{array}{cccc} & s_{0} & s_{1} & s_{2}\\ s_{0} & 0.5 & 0 & 0.5\\ s_{1} & 0.7 & 0.1 & 0.2\\ s_{2} & 0.4 & 0 & 0.6 \end{array}P_{1}\left(S,a_{1},S\right)=P_{0}\left(S,a_{1},S\right):\;\;\;\;\;\;\;\;\;\;\begin{array}{cccc} & s_{0} & s_{1} & s_{2}\\ s_{0} & 0 & 0 & 1\\ s_{1} & 0 & 0.95 & 0.05\\ s_{2} & 0.3 & 0.3 & 0.4 \end{array}$$
$$R_{1}\left(S,a_{0},S\right)=R_{0}\left(S,a_{0},S\right):\;\;\;\;\;\;\;\;\;\;\;\begin{array}{cccc} & s_{0} & s_{1} & s_{2}\\ s_{0} & 1 & 0 & 1\\ s_{1} & 5 & 1 & 1\\ s_{2} & 1 & 0 & 1 \end{array}R_{1}\left(S,a_{1},S\right)=R_{0}\left(S,a_{1},S\right):\;\;\;\;\;\;\;\;\;\;\;\begin{array}{cccc} & s_{0} & s_{1} & s_{2}\\ s_{0} & 0 & 0 & 1\\ s_{1} & 0 & 1 & 1\\ s_{2} & -1 & 1 & 1 \end{array}$$

There are 9 state groups in the CMDP system. Figure 3 depicts the change curve of the social value function under the strategy of $\left(\pi_{0},\pi_{1}\right)=\left(\begin{array}{ccc} s_{0} & s_{1} & s_{2}\\ a_{0} & a_{1} & a_{1} \end{array},\begin{array}{ccc} s_{0} & s_{1} & s_{2}\\ a_{1} & a_{0} & a_{1} \end{array}\right)$. As can be seen from the Figure 3, in the cooperation Markov decision process system introduced in this paper, as the number of iterations increases, the social value of the system increases and finally converges to a stable value.

In the CMDP system, agent Agent0 has NS0 states and agent Agent1 has NS1 states, where NS0=NS1. The set of actions that two agents can perform are {1,2,3,⋯,a} and each agent has an action that can be performed. Let *s* denote the state of agent Agent0, and *t* denote the state of agent Agent1, for a given strategy pair $\left(\pi_{0},\pi_{1}\right)$, let i=Pi0(s) denote the action performed by Agent0 in the *s* state under the $\pi_{0}$ strategy, let j=Pi0(t) denote the action perform by Agent1 in the *t* state under the $\pi_{1}$ strategy, among them, i,j∈{1,2,3,⋯,a}. The cooperation Markov decision process system with the participation of two agents performs iterative calculation according to Formula (18).

**Theorem** **3.**
*In a Cooperation Markov Decision Process system in which two agents participate, the two agents perform coordinated actions on the CMDP system, and the social value function converges.*


**Proof.** According to the iterative evolution process in Equation (Equation 18), we can get the calculation idea as follows:
(19)$$\begin{array}{c} \vec{V_{1}^{0}}=A_{1}+r\cdot P_{1}\cdot \vec{V_{0}},\vec{V_{1}^{1}}=A_{2}+r\cdot P_{2}\cdot \vec{V_{0}},\vec{V_{1}}=\alpha \vec{V_{1}^{0}}+\beta \vec{V_{1}^{1}};\\ \vec{V_{2}^{0}}=A_{1}+r\cdot P_{1}\cdot \vec{V_{1}},\vec{V_{2}^{1}}=A_{2}+r\cdot P_{2}\cdot \vec{V_{1}},\vec{V_{2}}=\alpha \vec{V_{2}^{0}}+\beta \vec{V_{2}^{1}};\\ \vec{V_{3}^{0}}=A_{1}+r\cdot P_{1}\cdot \vec{V_{2}},\vec{V_{3}^{1}}=A_{2}+r\cdot P_{2}\cdot \vec{V_{2}},\vec{V_{3}}=\alpha \vec{V_{3}^{0}}+\beta \vec{V_{3}^{1}};\\ \cdots \cdots \\ \vec{V_{k}^{0}}=A_{1}+r\cdot P_{1}\cdot \vec{V_{k-1}},\vec{V_{k}^{1}}=A_{2}+r\cdot P_{2}\cdot \vec{V_{k-1}},\vec{V_{k}}=\alpha \vec{V_{k}^{0}}+\beta \vec{V_{k}^{1}};\\ \vec{V_{k+1}^{0}}=A_{1}+r\cdot P_{1}\cdot \vec{V_{k}},\vec{V_{k+1}^{1}}=A_{2}+r\cdot P_{2}\cdot \vec{V_{k}},\vec{V_{k+1}}=\alpha \vec{V_{k+1}^{0}}+\beta \vec{V_{k+1}^{1}}. \end{array}$$

In Formula (19), $A_{1}$ and $P_{1}$ respectively represent the expected reward value matrices and probability transition matrix obtained by Agent0 under a certain state and a known strategy pair $\left(\pi_{0},\pi_{1}\right)$, among them, $A_{1}=diag\left(P_{1}•{R_{1}}^{T}\right)$ represents the probability of the state performing the action multiplied by the reward value for performing the action. $A_{2}$ and $P_{2}$ respectively represent the expected reward value matrix and probability transition matrix obtained by Agent0 under a certain state and a known strategy pair $\left(\pi_{0},\pi_{1}\right)$, among them, $A_{2}=diag\left(P_{2}•{R_{2}}^{T}\right)$ represents the probability of the state performing the action multiplied by the reward value for performing the action. The dimensions of $A_{1}$ and $A_{2}$ are NS0*NS1 rows and 1 column, *P* and *Q* are square matrices of NS0*NS1 dimension.
**Step** **1**Calculate $\vec{V_{1}}$. Given any initial value function $\vec{V_{0}}$ , it can be calculated using Equation (Equation 19),
(20)$$\begin{array}{c} \vec{V_{1}}=\alpha \vec{V_{1}^{0}}+\beta \vec{V_{1}^{1}}=\alpha \cdot \left(A_{1}+r\cdot P_{1}\cdot \vec{V_{0}}\right)+\beta \cdot \left(A_{2}+r\cdot P_{2}\cdot \vec{V_{0}}\right)\\ \;\;=\;\alpha A_{1}+\beta A_{2}+r\left(\alpha P_{1}+\beta P_{2}\right)\cdot \vec{V_{0}}. \end{array}$$**Step** **2**Find $\vec{V_{2}^{0}}$ and $\vec{V_{2}^{1}}$ according to $\vec{V_{1}}$ in Equation (Equation 19)
(21)$$\begin{array}{c} \vec{V_{2}^{0}}=A_{1}+r\cdot P_{1}\cdot \vec{V_{1}}=A_{1}+r\cdot P_{1}\cdot \left(\alpha A_{1}+\beta A_{2}+r\cdot \left(\alpha P_{1}+\beta P_{2}\right)\cdot \vec{V_{0}}\right)\\ \;\;\;=(1+r\alpha \cdot P_{1})A_{1}+\left(r\beta \cdot P_{1}\right)A_{2}+\left(r^{2}\alpha {P_{1}}^{2}+r^{2}\beta \cdot P_{1}\cdot P_{2}\right)\vec{V_{0}}, \end{array}$$
(22)$$\begin{array}{c} \vec{V_{2}^{1}}=A_{2}+r\cdot P_{2}\cdot \vec{V_{1}}=A_{2}+r\cdot P_{2}\cdot \left(\alpha A_{1}+\beta A_{2}+r\cdot \left(\alpha P_{1}+\beta P_{2}\right)\cdot \vec{V_{0}}\right)\\ \;\;\;=(r\alpha \cdot P_{2})A_{1}+\left(1+r\beta \cdot P_{2}\right)A_{2}+\left(r^{2}\alpha P_{2}\cdot P_{1}+r^{2}\beta {P_{2}}^{2}\right)\cdot \vec{V_{0}}. \end{array}$$

It can be known from Equation (Equation 19) that by substituting $\vec{V_{2}^{0}}$ in Equation (Equation 21) and $\vec{V_{2}^{1}}$ in Equation (22) into $\vec{V_{2}}$, the expression of $\vec{V_{3}}$ can be derived as follows:
(23)$$\begin{array}{c} \vec{V_{2}}=\alpha \vec{V_{2}^{0}}+\beta \vec{V_{2}^{1}}\\ \;\;=\;\alpha \cdot [(1+r\alpha \cdot P_{1})A_{1}+\left(r\beta \cdot P_{1}\right)A_{2}+\left(r^{2}\alpha {P_{1}}^{2}+r^{2}\beta \cdot P_{1}\cdot P_{2}\right)\vec{V_{0}}]+\\ \;\;\beta \cdot \left[(r\alpha \cdot P_{2}\right)A_{1}+\left(1+r\beta \cdot P_{2}\right)A_{2}+\left(r^{2}\alpha P_{2}\cdot P_{1}+r^{2}\beta {P_{2}}^{2}\right)\cdot \vec{V_{0}}]\\ \;\;=(\alpha +r\alpha^{2}\cdot P_{1}+r\alpha \beta \cdot P_{2})A_{1}+\left(\beta +\alpha r\beta \cdot P_{1}+r\beta^{2}\cdot P_{2}\right)A_{2}+\\ \;\;\left(r^{2}\alpha^{2}{P_{1}}^{2}+r^{2}\alpha \beta \cdot P_{1}\cdot P_{2}+r^{2}\alpha \beta P_{2}\cdot P_{1}+r^{2}\beta^{2}{P_{2}}^{2}\right)\cdot \vec{V_{0}}\\ \;\;=[\alpha +r\alpha \left(\alpha P_{1}+\beta P_{2}\right)A_{1}]+[\beta +r\beta \left(\alpha P_{1}+\beta P_{2}\right]A_{2}+r^{2}{\left(\alpha P_{1}+\beta P_{2}\right)}^{2}\cdot \vec{V_{0}}. \end{array}$$
**Step** **3**According to Equation (Equation 19), $\vec{V_{2}}$ can be used to find $\vec{V_{3}^{0}}$ and $\vec{V_{3}^{1}}$
(24)$$\begin{array}{c} \vec{V_{3}^{0}}=A_{1}+r\cdot P_{1}\cdot \vec{V_{2}}\\ \;\;\;=A_{1}+rP_{1}\cdot \left\{\alpha +r\alpha \left(\alpha P_{1}+\beta P_{2}\right)A_{1}]+[\beta +r\beta \left(\alpha P_{1}+\beta P_{2}\right]A_{2}+r^{2}{\left(\alpha P_{1}+\beta P_{2}\right)}^{2}\cdot \vec{V_{0}}\right\}\\ \;\;\;=(1+r\alpha P_{1}+r^{2}\alpha^{2}\cdot {P_{1}}^{2}+r^{2}\alpha \beta P_{1}\cdot P_{2})A_{1}+\left(r\beta P_{1}+\alpha r^{2}\beta \cdot {P_{1}}^{2}+r^{2}\beta^{2}P_{1}\cdot P_{2}\right)A_{2}\\ \;\;\;+r^{3}\cdot P_{1}\cdot {\left(\alpha P_{1}+\beta P_{2}\right)}^{2}\cdot \vec{V_{0}}, \end{array}$$
(25)$$\begin{array}{c} \vec{V_{3}^{1}}=A_{2}+r\cdot P_{2}\cdot \vec{V_{2}}\\ \;\;\;=A_{2}+r\cdot P_{2}\cdot \left[(\alpha +r\alpha^{2}P_{1}+r\alpha \beta P_{2}\right)A_{1}+\left(\beta +\alpha r\beta P_{1}+r\beta^{2}P_{2}\right)A_{2}+r^{2}{\left(\alpha P_{1}+\beta P_{2}\right)}^{2}\cdot \vec{V_{0}}]\\ \;\;\;=(r\alpha P_{2}+r^{2}\alpha^{2}P_{2}\cdot P_{1}+r^{2}\alpha \beta \cdot {P_{2}}^{2})A_{1}+\left(1+r\beta P_{2}+\alpha r^{2}\beta P_{2}\cdot P_{1}+r^{2}\beta^{2}\cdot {P_{2}}^{2}\right)A_{2})\\ \;\;\;+r^{3}\cdot P_{2}{\left(\alpha P_{1}+\beta P_{2}\right)}^{2}\cdot \vec{V_{0}}. \end{array}$$

We can known from Equation (Equation 19) that by substituting $\vec{V_{3}^{0}}$ in Equation (Equation 24) and $\vec{V_{3}^{1}}$ in Equation (25) into $\vec{V_{3}}$, the expression of $\vec{V_{3}}$ can be further obtained
(26)$$\begin{array}{c} \vec{V_{3}}=\alpha \vec{V_{3}^{0}}+\beta \vec{V_{3}^{1}}\\ \;\;\;=\alpha \left[(1+r\alpha P_{1}+r^{2}\alpha^{2}\cdot {P_{1}}^{2}+r^{2}\alpha \beta P_{1}\cdot P_{2}\right)A_{1}+\left(r\beta P_{1}+\alpha r^{2}\beta \cdot {P_{1}}^{2}+r^{2}\beta^{2}P_{1}\cdot P_{2}\right)A_{2}+\\ \;\;\;r^{3}\cdot P_{1}\cdot {\left(\alpha P_{1}+\beta P_{2}\right)}^{2}\cdot \vec{V_{0}}]+\beta \left[(r\alpha P_{2}+r^{2}\alpha^{2}P_{2}\cdot P_{1}+r^{2}\alpha \beta \cdot {P_{2}}^{2}\right)A_{1}+\\ \;\;\;\left(1+r\beta P_{2}+\alpha r^{2}\beta P_{2}\cdot P_{1}+r^{2}\beta^{2}\cdot {P_{2}}^{2}\right)A_{2})+r^{3}\cdot P_{2}{\left(\alpha P_{1}+\beta P_{2}\right)}^{2}\cdot \vec{V_{0}}]\\ \;\;\;=(\alpha +r\alpha^{2}P_{1}+r^{2}\alpha^{3}\cdot {P_{1}}^{2}+r^{2}\alpha^{2}\beta P_{1}\cdot P_{2}+r\alpha \beta P_{2}+r^{2}\alpha^{2}\beta P_{2}\cdot P_{1}+r^{2}\alpha \beta^{2}\cdot {P_{2}}^{2})A_{1}+\\ \;\;\;(r\alpha \beta P_{1}+\alpha^{2}r^{2}\beta \cdot {P_{1}}^{2}+r^{2}\alpha \beta^{2}P_{1}\cdot P_{2}+\beta +r\beta^{2}P_{2}+\alpha r^{2}\beta^{2}P_{2}\cdot P_{1}+r^{2}\beta^{3}\cdot {P_{2}}^{2})A_{2}+\\ \;\;\;[\alpha r^{3}\cdot P_{1}\cdot {\left(\alpha P_{1}+\beta P_{2}\right)}^{2}+\beta r^{3}\cdot P_{2}{\left(\alpha P_{1}+\beta P_{2}\right)}^{2}]\cdot \vec{V_{0}}\\ \;\;\;=[\alpha +r\alpha \left(\alpha P_{1}+\beta P_{2}\right)+r^{2}\alpha {\left(\alpha P_{1}+\beta P_{2}\right)}^{2}]A_{1}+[\beta +r\beta \left(\alpha P_{1}+\beta P_{2}\right)+r^{2}\beta {\left(\alpha P_{1}+\beta P_{2}\right)}^{2}]\cdot \\ \;\;\;A_{2}+r^{3}{\left(\alpha P_{1}+\beta P_{2}\right)}^{3}\cdot \vec{V_{0}}. \end{array}$$

Finally, from the expressions of $\vec{V_{1}}$, $\vec{V_{2}}$ and $\vec{V_{3}}$, it is not difficult to find the laws through observation. The value function $\vec{V_{k}}$ at time *k* and the value function $\vec{V_{k+1}}$ at time k+1 can be deduced by the following Formulas (27) and (28) as shown.
(27)$$\begin{array}{c} \vec{V_{k}}=\alpha \vec{V_{k}^{0}}+\beta \vec{V_{k}^{1}}\\ \;\;=[\alpha +r\alpha \left(\alpha P_{1}+\beta P_{2}\right)+r^{2}\alpha {\left(\alpha P_{1}+\beta P_{2}\right)}^{2}+\cdots +r^{k-1}\alpha {\left(\alpha P_{1}+\beta P_{2}\right)}^{k-1}]A_{1}+\\ \;\;\;[\beta +r\beta \left(\alpha P_{1}+\beta P_{2}\right)+r^{2}\beta {\left(\alpha P_{1}+\beta P_{2}\right)}^{2}+\cdots +r^{k-1}\beta {\left(\alpha P_{1}+\beta P_{2}\right)}^{k-1}]A_{2}+\\ \;\;\;r^{k}{\left(\alpha P_{1}+\beta P_{2}\right)}^{k}\cdot \vec{V_{0}}, \end{array}$$
(28)$$\begin{array}{c} \vec{V_{k+1}}=\alpha \vec{V_{k+1}^{0}}+\beta \vec{V_{k+1}^{1}}\\ \;\;=[\alpha +r\alpha \left(\alpha P_{1}+\beta P_{2}\right)+\cdots +r^{k-1}\alpha {\left(\alpha P_{1}+\beta P_{2}\right)}^{k-1}+r^{k}\alpha {\left(\alpha P_{1}+\beta P_{2}\right)}^{k}]A_{1}+\\ \;\;\;[\beta +r\beta \left(\alpha P_{1}+\beta P_{2}\right)++\cdots +r^{k-1}\beta {\left(\alpha P_{1}+\beta P_{2}\right)}^{k-1}+r^{k}\beta {\left(\alpha P_{1}+\beta P_{2}\right)}^{k}]A_{2}+\\ \;\;\;r^{k+1}{\left(\alpha P_{1}+\beta P_{2}\right)}^{k+1}\cdot \vec{V_{0}}. \end{array}$$

Subtract the two equations to calculate the value function difference between time *k* and k+1:
$$\begin{array}{c} \vec{V_{k+1}}-\vec{V_{k}}\\ =[r^{k}\alpha {\left(\alpha P_{1}+\beta P_{2}\right)}^{k}]A_{1}+r^{k}\beta {\left(\alpha P_{1}+\beta P_{2}\right)}^{k}]A_{2}+r^{k+1}{\left(\alpha P_{1}+\beta P_{2}\right)}^{k+1}\cdot \vec{V_{0}}-r^{k}{\left(\alpha P_{1}+\beta P_{2}\right)}^{k}\cdot \vec{V_{0}}\\ =[r^{k}\alpha {\left(\alpha P_{1}+\beta P_{2}\right)}^{k}]A_{1}+r^{k}\beta {\left(\alpha P_{1}+\beta P_{2}\right)}^{k}]A_{2}+\left[r^{k}{\left(\alpha P_{1}+\beta P_{2}\right)}^{k}\cdot \left(r\left(\alpha P_{1}+\beta P_{2}\right)-E\right)\right]\cdot \vec{V_{0}}\\ =r^{k}{\left(\alpha P_{1}+\beta P_{2}\right)}^{k}\cdot \left\{\alpha A_{1}+\beta A_{2}+\left[r\left(\alpha P_{1}+\beta P_{2}\right)-E\right]\cdot \vec{V_{0}}\right\}. \end{array}$$

$P_{1}$ and $P_{2}$ are two probability transition matrices, α+β=1, thus, $\alpha P_{1}+\beta P_{2}$ is a probability transition matrix and the *k*-th power of the probability transition matrix is still the probability matrix, so ${\left(\alpha P_{1}+\beta P_{2}\right)}^{k}$ is a probability transition matrix. Because of discount factor 0<r<1, then $r^{k}$ decreases monotonically with the increase of *k*. When *k* is large enough, $r^{k}\to 0$. That is to say:
$$\lim_{k\to \infty }\vec{(V_{k+1}}-\vec{V_{k}})=\lim_{k\to \infty }r^{k}{\left(\alpha P_{1}+\beta P_{2}\right)}^{k}\cdot \left\{\alpha A_{1}+\beta A_{2}+\left[r\left(\alpha P_{1}+\beta P_{2}\right)-E\right]\cdot \vec{V_{0}}\right\}=0.$$

So, for any given strategy pair $\left(\pi_{0},\pi_{1}\right)$, $\vec{V_{k+1}}=\vec{V_{k}}$, in the CMDP system introduced in this paper, the final value function *V* will converge. If the discount factor is larger, the convergence speed is slower, the smaller the discount factor is, the faster the convergence speed is. □

**Theorem** **4.**
*In the Cooperation Markov Decision Process system, the final value $\vec{V_{k}}$ of the value function is independent of the initial value of $\vec{V_{0}}$.*


**Proof.** From Theorem 3, the value function at time can be expressed, and the expression (27) can be further simplified to obtain:
$$\begin{array}{c} \vec{V_{k}}=\alpha \vec{V_{k}^{0}}+\beta \vec{V_{k}^{1}}\\ \;\;=[\alpha +r\alpha \left(\alpha P_{1}+\beta P_{2}\right)+r^{2}\alpha {\left(\alpha P_{1}+\beta P_{2}\right)}^{2}+\cdots +r^{k-1}\alpha {\left(\alpha P_{1}+\beta P_{2}\right)}^{k-1}]A_{1}+\\ \;\;\;[\beta +r\beta \left(\alpha P_{1}+\beta P_{2}\right)+r^{2}\beta {\left(\alpha P_{1}+\beta P_{2}\right)}^{2}+\cdots +r^{k-1}\beta {\left(\alpha P_{1}+\beta P_{2}\right)}^{k-1}]A_{2}+r^{k}{\left(\alpha P_{1}+\beta P_{2}\right)}^{k}\cdot \vec{V_{0}}\\ \;\;=(\alpha A_{1}+\beta A_{2})+\left[r\left(\alpha P_{1}+\beta P_{2}\right)\cdot \left(\alpha A_{1}+\beta A_{2}\right)\right]+\left[r^{2}{\left(\alpha P_{1}+\beta P_{2}\right)}^{2}\cdot \left(\alpha A_{1}+\beta A_{2}\right)\right]+\\ \;\;\;+\cdots +\left[r^{k-1}{\left(\alpha P_{1}+\beta P_{2}\right)}^{k-1}\cdot \left(\alpha A_{1}+\beta A_{2}\right)\right]+r^{k}{\left(\alpha P_{1}+\beta P_{2}\right)}^{k}\cdot \vec{V_{0}}\\ \;=\left[E+r\left(\alpha P_{1}+\beta P_{2}\right)+r^{2}{\left(\alpha P_{1}+\beta P_{2}\right)}^{2}+\cdots +r^{k-1}{\left(\alpha P_{1}+\beta P_{2}\right)}^{k-1}\right]\cdot (\alpha A_{1}+\beta A_{2})+\\ \;\;\;r^{k}{\left(\alpha P_{1}+\beta P_{2}\right)}^{k}\cdot \vec{V_{0}}. \end{array}$$Obviously, $P_{1}$ and $P_{2}$ are two probability transition matrices, α+β=1, thus $\left(\alpha P_{1}+\beta P_{2}\right)$ is a probability transition matrix, ∀k∈N, ${\left(\alpha P_{1}+\beta P_{2}\right)}^{k}$ is a probability transition matrix. *r* is the discount factor and 0<r<1, then with the increase of *k*, the $r^{k}$ decreases monotonically. When *k* is large enough, $r^{k}\to 0$, that $\lim_{k\to \infty }r^{k}=0$, so $\lim_{k\to \infty }r^{k}{\left(\alpha P_{1}+\beta P_{2}\right)}^{k}\cdot \vec{V_{0}}=\vec{0}$, we can conclude,
$$\begin{array}{c} \lim_{k\to \infty }\vec{V_{k}}=\lim_{k\to \infty }\left[E+r\left(\alpha P_{1}+\beta P_{2}\right)+r^{2}{\left(\alpha P_{1}+\beta P_{2}\right)}^{2}+\cdots +r^{k-1}{\left(\alpha P_{1}+\beta P_{2}\right)}^{k-1}\right]\cdot (\alpha A_{1}+\beta A_{2})+\\ \;\;\;r^{k}{\left(\alpha P_{1}+\beta P_{2}\right)}^{k}\cdot \vec{V_{0}}\\ \;\;=\lim_{k\to \infty }\left[E+r\left(\alpha P_{1}+\beta P_{2}\right)+r^{2}{\left(\alpha P_{1}+\beta P_{2}\right)}^{2}+\cdots +r^{k-1}{\left(\alpha P_{1}+\beta P_{2}\right)}^{k-1}\right]\cdot (\alpha A_{1}+\beta A_{2}). \end{array}$$Therefore, the choice of the initial function $\vec{V_{k}}$ does not affect the convergence value of the final value function $\vec{V_{0}}$. □

### 4.3. Algorithm for Optimal Strategy Pairs in Cooperation Type CMDP System

From the theoretical results in Section 4.2, it can be known that under the strategy pair $\left(\pi_{0},\pi_{1}\right)$, the *V*-function generated by the iteration of Equation (Equation 18) is recorded as $V^{\left(\pi_{0},\pi_{1}\right)}$ to form a stable value. Similarly, according function *V* and improve $\left(\pi_{0},\pi_{1}\right)$ to obtain $\left(\pi_{0}^{{}^{\prime }},\pi_{1}^{{}^{\prime }}\right)$ by the following greedy method.
(29)$$\begin{array}{c} \pi_{0}^{{}^{\prime }}\left(s,t\right)=\arg\max_{a}\Sigma_{(s^{\prime },a^{\prime })\in S\times A}^{}P_{0}\left(s,t,a,s^{\prime },a^{\prime }\right)\left[R_{0}\left(s,t,a,s^{\prime },a^{\prime }\right)+\gamma \cdot V_{}^{\left(\pi_{0}\;,\;\;\pi_{1}\right)}\left(s^{\prime },t\right)\right]\;\;\;\\ \pi_{1}^{{}^{\prime }}\left(s,t\right)=\arg\max_{a}\Sigma_{(t^{\prime },a^{\prime })\in S\times A}^{}P_{1}\left(s,t,a,t^{\prime },a^{\prime }\right)\left[R_{1}\left(s,t,a,t^{\prime },a^{\prime }\right)+\gamma \cdot V_{}^{\left(\pi_{0}\;,\;\;\pi_{1}\right)}\left(s,t^{\prime }\right)\right]\;\;. \end{array}$$

In order to obtain the algorithm of the optimal strategy pair $\left(\pi_{0}^{*},\pi_{1}^{*}\right)$ in the Cooperation Markov Decision Process system. We modeled the optimal strategy of a single agent. Let Formula (18) be the iterative calculation formula of the “evolution module” part of the algorithm, Formula (29) be the strategy improvement method of the “strategy improvement module” in the algorithm. According the method of solving the single-agent Markov decision process system’s optimal strategy, this section presents the algorithm for solving the optimal strategy of the joint type cooperation Markov decision process system, the optimal strategy pair is $\left(\pi_{0}^{*},\pi_{1}^{*}\right)$.

The detail code of Algorithm 1 can see in Appendix A. The execution process of the above algorithm is shown in the following Figure 4:
**Algorithm 1** The Optimal Strategy Pairs**Input:** input CMDP quintuple <S,A,{Agent0,Agent1},{T0,T1},{R0,R1}>: error parameter, ε>0 balance parameter 0<α<1, discount factor 0<γ<1.**Output:** output Optimal strategy pair (π*0,π*1)=(πk0,πk1), optimal value function Vk+1=Vk+1(π*0,π*1).Initialization *V*-function V10,V11, take the initial value randomly. Such as V11(s)=0(s∈S); suppose V1=αV10+(1−α)V11;Use Equation (Equation 19) to greedy improvement strategy pair (π10,π11): (V1(π10,π11)=V1);Use the updated strategy pair (π10,π11) in step 2 and Formula (18) to find the *V*-function V2;Repeat steps 2, 3;Step k+1: assuming (πk0,πk1), Vk(πk0,πk1) has been obtained, at this step, do the following two steps of calculation:
(a)Evolutionary calculation of CMDP system: find Vk+1 by (πk0,πk1) and Formula (18), when ∥Vk+1−Vk∥<ε, defined Vk+1(πk0,πk1)(s,t)=Vk(s,t)((s,t)∈S×S).(b)Greedy computing (πk+10,πk+11):π0,k+1(s,t)=argmaxaΣ(s′,a′)∈S×AP0(s,t,a,s′,a′)[R0(s,t,a,s′,a′)+γ·V0(π0k,π1k)(s′,t)],π1,k+1(s,t)=argmaxaΣ(t′,a′)∈S×AP1(s,t,a,t′,a′)[R1(s,t,a,t′,a′)+γ·V0(π0k,π1k)(s,t′)].If πk+10=πk0, πk+11=πk1, terminate calculation.**Return** result.

## 5. An Application Example of Cooperation Markov Decision Process System

Reinforcement learning is one of the most concerned topic in artificial intelligence after deep learning. Reinforcement learning agents interact with the environment and take actions that receive feedback similar to learning strategies in medicine. In fact, many applications of reinforcement learning in health care have to do with finding the best treatment strategy. Many reading learning software also use intensive learning to provide customized learning materials and content for the tutoring system to meet the specific needs of students. The recommendation system of e-commerce changes the recommendation strategy in real time according to the feedback of users (ignore, click and buy).

The core problem of multi-agent system research is to seek to establish an effective cooperation mechanism, so that multi-agent with simple function and independent function can complete complex target tasks or solve complex problems through negotiation, coordination and cooperation. In the multi-agent system, each agent learns and improves its own strategy by interacting with the environment to obtain the reward value, so as to obtain the optimal strategy under the environment is the process of multi-agent reinforcement learning. The transfer of multi-agent state depends on the actions of all agents, and the joint actions determine the transfer of the whole system. The return of each agent is not only related to its own actions, but also the actions of other agents will affect the return of the current agent. If multi-agent are involved, there are three types of cooperative game, competition game and mixed game combined between agents in the same environment.

The proposed joint multi-agent in this paper only analyzes the relationship between two agents, two agents take interactive actions and complete cooperation to achieve the goal together. When intelligence agent *B* observed intelligence agent *A* take an action, intelligence agent *B* will perform an action. At the same time, the next action of agent *A* also depends on the agent *B*’s action. Repeat the process of perform action above, the behavior of the two agents influence each other and evolve and learn in the CMDP system of a given strategy pair to find a pair of optimal strategies finally.

Background: a couple found a gold bar not far from home at the same time, this gold bar needs two people together, one carried one end to carry home, they want to carry the gold bar to home, they need avoid the trap and then everyone arrives at the side of the gold bar, after lifting the gold bar, they also have to bypass the obstacles in front of the home, finally, the gold bar is successfully transported home.

The background is transformed into the environment as shown in the Figure 5, the simulation is carried out on the figure. Two people are compared to two agents respectively. In addition, the figure can only move up, down, or left, and each position can only accommodate one agent. In this experiment, agents in the same environment, through mutual cooperation, the gold bars to the designated location. The whole process is divided into the following three steps:**Step** **1**Bypass the trap. When passing through the trap area, each position can only accommodate one agent, two agents cannot be in the same position at the same time, so they need to interact with each other which agent passe first. When agent *B* observes that agent *A* is going to pass through the trap position, agent *B* cannot pass through the trap position.**Step** **2**Reach the designated position. The two agents must reach the two ends of the gold bar to pick up the gold bar, so the two agents can not reach the same place at the same time. When one of the agents reaches one end of the gold bar, the other must walk to the other end of the gold bar before the two agents can work together to pick up the gold bar.**Step** **3**Delivery of gold bar. Because the gold bar has a certain length, in the process of transporting the gold bar to the designated position, the two agents are at different positions. In the process of transporting the gold bar, the two agents also need to consider how to bypass the obstacles and the positions of the two agents. The two agents need to cooperate and interact to complete the task.

In the process of gold bar transportation, through continuous exploration and trial and error in the environment, the two intelligence agent will find the best way to move. Finally, the two agents cooperate through the trap to reach the two ends of the gold bar, and then bypass the obstacles to successfully transport the gold bar home. In such an example, each agent can only move up, down, left, and right to adjacent positions in the environment, and the sum of the probabilities of moving to all adjacent positions is equal to 1. The agent obtains certain benefits by moving the position. In order to avoid falling into traps and avoiding obstacles, the probability and benefits of moving to the vicinity of traps and obstacles are also relatively small compared to other locations. The goal is that through constant trial and error and learning in the environment, the two agents can take an optimal way of movement to get the most benefit. We express the probability and reward value of moving adjacent positions in this problem with a square matrix, and the probability and reward value of the joint action of two agents are the tensor product of two probability matrices and two reward matrices. Using the algorithm 1 in this article, given an initial strategy, the system will obtain a value function under this strategy. Under the value function at this time, the agent performs all joint actions and selects the joint action that enables the system to obtain the highest reward value. Use this to modify the initial strategy, and then obtain a new value function under the modified strategy. According to this method, iteratively modify the strategy and value function, and finally evolve to learn a strategy that can obtain the maximum reward value. That is, the optimal strategy.

Conversely, if the two agents do not cooperate in any of the steps described in the appeal, it is difficult to successfully transport the gold bar home. Firstly, there will be a conflict when they bypass the trap, both agents cannot simultaneously adopt their own optimal way to pass the trap position. Secondly, after passing through the trap, then something happens that the two agents are on the same side of the bar. After that, the two agents may not move in the same direction, which will also cause the task to fail. So in such an instance, the interaction between two agents becomes extremely important.

## 6. Conclusions

This paper first proves the convergence of the value function of the monomer Markov decision process system. Furthermore, a cooperation Markov decision process system of two agents is introduced, which is suitable for an evolutionary learning system of cooperative (or antagonistic) decision between two agents. The main difference between cooperation and confrontation lies in the way the value function is defined and the way of combination (or restriction). In this paper, only CMDP system of cooperative type is considered. In this kind of learning model, agents alternately execute behaviors, seek the optimal criterion with social value, find the optimal strategy $\left(\pi_{0}^{*},\pi_{1}^{*}\right)$, and jointly complete the target task. This paper presents an algorithm for finding the optimal strategy pair $\left(\pi_{0}^{*},\pi_{1}^{*}\right)$, whose fundamental task is to find a pair of optimal strategies to form an evolutionary system $CMDP^{\;(\pi_{0}^{*},\pi_{1}^{*})}$. Finally, the convergence of the value function in system CMDP is proved, and the convergence value is not affected by the value of the initial value function. This paper only introduces the cooperation Markov decision process system involving two agents, but many problems in reality involve more agents. As the number of agents increases, the interaction between agents in the system becomes more complex, and the difficulty of analysis also increases exponentially. Based on the fact that deep learning has strong perceptual ability, but lacks certain decision-making ability, while reinforcement learning has decision-making ability but short of doing anything about perceptual problems. Therefore, combining the two and complementing each other’s advantages, a new concept–deep reinforcement learning is proposed. so we will use the next deep reinforcement learning method to study the properties of the cooperation Markov decision process system.

## Figures and Tables

**Figure 1 entropy-22-00955-f001:**
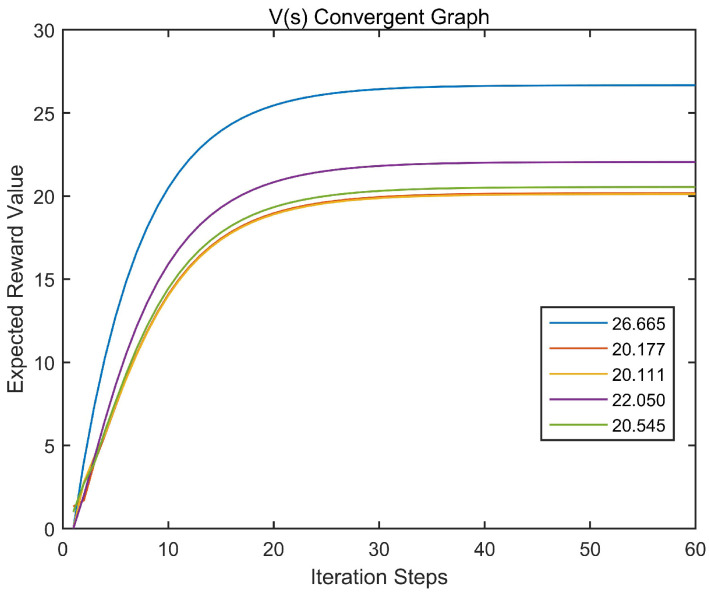
*V*-function convergence diagram.

**Figure 2 entropy-22-00955-f002:**
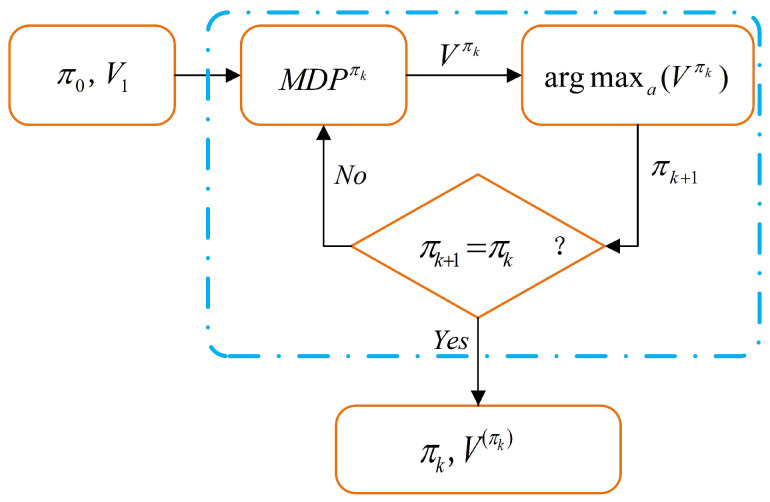
Optimal strategy solution algorithm framework.

**Figure 3 entropy-22-00955-f003:**
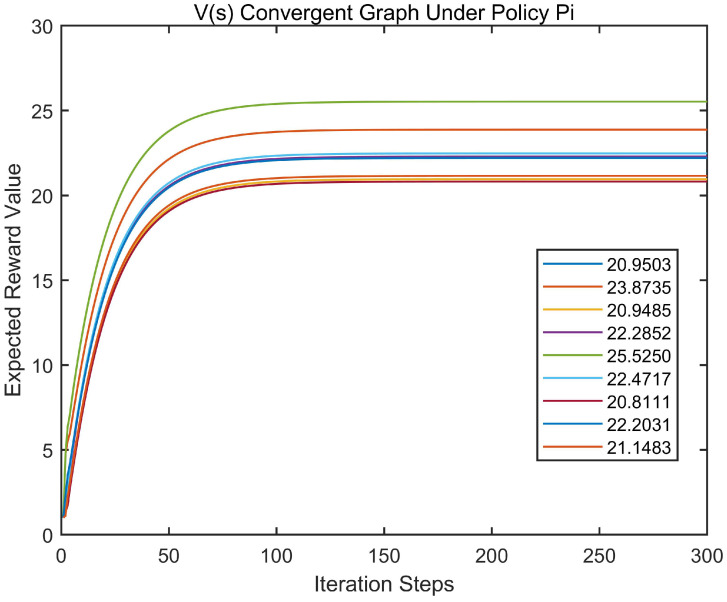
Convergence diagram of the social value function of the Cooperation Markov Decision Process (CMDP) system.

**Figure 4 entropy-22-00955-f004:**
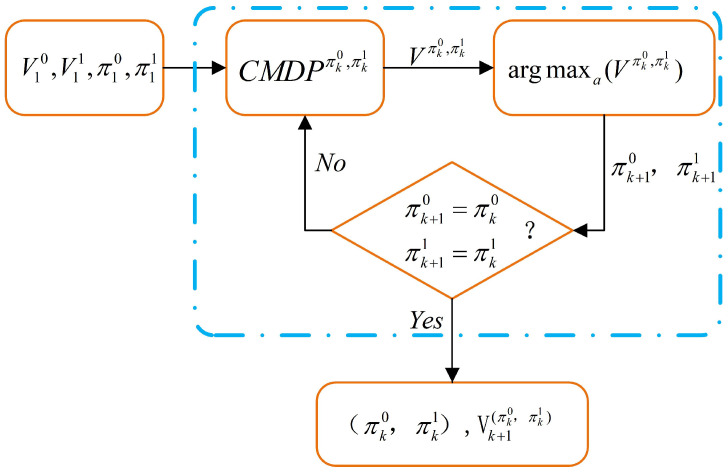
Optimal Strategy Solution Algorithm Framework of CMDP System.

**Figure 5 entropy-22-00955-f005:**
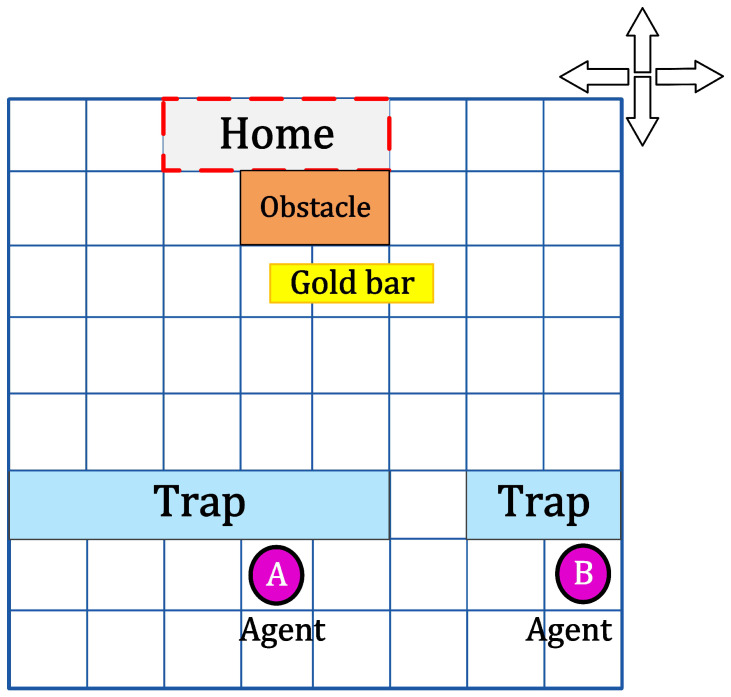
The environment of two agents in this example.

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
