# Peer review of "The Convergence of a Cooperation Markov Decision Process System"

_entropy, 2020, doi:10.3390/e22090955_

Round 1

Reviewer 1 Report

Dear Authors,

The submitted paper " The Convergence of a Cooperation Markov Decision Process System" is addressing an important and interesting topic, therefore thank you very much for your work and the contribution. The Authors propose the convergence of the value function of the monomer Markov decision system. Additionally, a Cooperation Markov decision system (CMDPs) of two agents is introduced, which is suitable for an evolutionary learning system of cooperative (or antagonistic) decision between two agents.. The provided example exploited the applicability of the proposed approach.

Generally, paper is well structured, important theoretical aspects of the examined  problem are studied and presented in a clear and consistent manner.  According reviewer best knowledge I found the proposed approach new. In my opinion the subject of the paper is directly related to the Journal’s main topics.

Consequently, the paper should be revised based on the comments of  the reviewer, which are cited below:

- literature review should be provided

- research gap should be highlighted in the light of the up to date relevant studies (lack of up to date references)

- findings and contributions should be clearly identified and highlighted

- while usefulness of Authors’ approach is presented in case study, the manuscript lacks of competition of Authors approach with other methods

- please focus on technical quality and language layer of manuscript (what is the word “systemy”??)

Reviewer 2 Report

Manuscript ID: entropy-893121

Title: The Convergence of a Cooperation Markov Decision Process System

This paper develops a Cooperative Markov Decision Process system. It is an interesting topic. I believe it will find a wide range of application field from all industries. However, a major revision is required based on the following concerns.

The sections 2, 3 and 4 deal with a mass of formulas. Finally, the authors propose their model in Figure 4 which is the developed version of Figure 2. This is what the authors do, and I see that there exist some proofs in these sections. However, there is a big problem. In section 5, the authors provide an example and they explain a problem. So, they did not make an application. They did not solve the problem. It is just an explanation of a problem and how to solve it. Did you solve that problem? No. Where is your algorithm or codes or your solution? I want to see it; I want to implement it; I want to run and check it if your approach is correct or not. Otherwise, I can produce tens of similar problems and thus different algorithms. It might be very interesting for other industries (i.e. ships passing a channel, ship navigation in the same environment) if you can provide the understandable codes or solutions that everyone can implement. It should be generalized and applied to all kinds of problems from different disciplines.

Abbreviations of the paper should be checked throughout the whole paper. They should be placed where they are firstly mentioned.

There exist some typos, some errors of capitals-lower-case letters (i.e. machine Learning) and blanks (from state sto state)

The concepts and the terms in the paper should be consistent. It might be confusing for the readers. For example, Cooperative Markov Decision Process system vs Cooperative Markov system or CMDPs vs CMDP, etc.

The paper should be checked, and the problematic sentences should be corrected. There are some problems in those sentences. (i.e. Markov decision system the main goal of the decision system is to find an optimal strategy – intelligence individual- of the search for the optimal strategy for.- oberved that- only one agent is allow to - systemy) ... English of the paper should be edited.

A major revision is required.

Round 2

Reviewer 1 Report

please improve technical quality and layout of figure 5

Reviewer 2 Report

I think this is better now.

Please remember to link the supplementary files with the manuscript

I believe the revision is complete and it is ready to get published. 
